# The Association of Triiodothyronine-to-Thyroxine Ratio with Body Mass Index in Obese Nigerian Children and Adolescents

**DOI:** 10.3390/medsci5040036

**Published:** 2017-12-15

**Authors:** Mathias Abiodun Emokpae, Progress Arhenrhen Obazelu

**Affiliations:** Department of Medical Laboratory Science, School of Basic Medical Sciences, College of Medical Sciences, University of Benin, Benin City 300001, Nigeria; progress.obazelu@uniben.edu

**Keywords:** obesity, thyroid hormones, children, adolescents

## Abstract

The interest in the relationship between thyroid dysfunction and obesity is on the increase. This study compares the triiodothyronine-to-thyroxine (T3/T4) ratio in obese and lean children and adolescents, and correlates thyroid hormones with body mass index (BMI) in obese Nigerian children. It is a retrospective study of records of 76 obese children and adolescents with a BMI of 31.7 ± 0.1 kg/m^2^ (26 males aged 10.9 ± 0.35 years, and 50 females aged 10.8 ± 0.4 years) that were referred to the laboratory for thyroid hormone evaluation because of their obese status. The controls were 20 age-matched non-obese apparently healthy subjects, with a mean age of 11.0 ± 0.47 years and a BMI of 20.2 ± 0.2 kg/m^2^. Serum T3, T4, and thyroid stimulating hormone (TSH) were determined using ELECSYS 1010 auto-analyzer (Roche Diagnostics, Penzberg, Germany). The BMI (*p* < 0.001), T3 (*p* < 0.01), TSH (*p* < 0.001) and T3/T4 ratio (*p* < 0.001) were significantly higher in obese than non-obese children and adolescents. Triiodothyronine (*r* = 0.230; *p* < 0.05), TSH (*r* = 0.272; *p* < 0.02), and T3/T4 ratio (*r* = 0.232; *p* < 0.05) correlated positively with BMI in obese children and adolescents. The T3/T4 ratio (*p* < 0.005) was significantly higher in obese boys than obese girls. Serum T3, TSH, and T3/T4 ratio correlated positive with BMI in obese Nigerian children and adolescents. Since thyroid dysfunction represents a continuum from asymptomatic to clinical symptomatic disease, it is suggested that obese children be counseled on the need to maintain ideal BMI in order to avoid the risks associated with obesity.

## 1. Introduction

Childhood obesity has emerged as a public health challenge in both developed and developing countries of the world. The interest in the relationship between thyroid dysfunction and obesity is on the increase. Obesity is defined as body mass index (BMI) of ≥95th percentile for children and adolescents of the same age and gender. The thyroid hormones are involved in the control of energy homeostasis and impact on body weight, thermogenesis, and lipid metabolism [1,2]. In addition, it was reported that thyroid stimulating hormone (TSH) induces differentiation of pre-adipocytes into adipocytes via its receptors in fat tissues (adipocytes) [3,4]. Adipocytes also secrete leptin, which acts on the central nervous system to modulate the neuroendocrine activity of the hypothalamus and pituitary [1]. Thyroid function has been extensively studied elsewhere in obese adults [5,6,7,8,9,10,11], but there appears to be a paucity of literature on childhood obesity in our setting. Previous studies have reported high serum levels of TSH with normal levels of triiodothyronine (T3) and thyroxine (T4) (sub-clinical hypothyroidism) in obese subjects [7,8,9,10]. Thyroid abnormalities were previously regarded as a cause of obesity, but recent studies have observed that they are considered an adaptation process aimed at increasing resting energy expenditure and total energy expenditure. That excess weight can impact on thyroid function, with the presence of hyperthyrotropinemia, with or without changes in T3 and T4 levels. It was therefore suggested that affected subjects may not require any medical treatment because thyroid hormones do normalize following substantial weight loss [1].

According to a World Health Organization report of 2013, the number of overweight and obese children under the age of five years was estimated to be 42 million [12]. The pathophysiology of overweight and obese subjects includes behavioral, environmental, metabolic, genetic, and psychological factors [13,14]. Some of the disorders associated with obesity are dyslipidemia, hypertension, impaired glucose tolerance, insulin resistance, and non-alcoholic steatohepatitis. These disorders predispose obese subjects to increased risk of cardiovascular diseases and diabetes.

We previously reported sub-clinical hypothyroidism in Nigerian obese children and adolescents characterized by elevated TSH [15]. Elevated TSH may be due to iodine deficiency, auto-immune thyroiditis, or non-synonymous mutations of the TSH gene [16]. Some authors have shown a positive association between TSH and BMI in Asians [17] and European countries [18], but there appears to be an inconsistency in the association of thyroid hormones with obesity [19,20]. There also appears to be a paucity of report on the relationship between T3/T4 ratio and BMI in obese children and adolescents in Nigeria. The T3/T4 ratio is a calculated index used to indicate thyroid function and the action of hormones on the tissue. It is vital in the diagnosis of borderline thyrotoxicosis, euthyroid sick syndrome, treatment monitoring during thyroid hormones replacement therapy, and iodine deficiency [21,22]. It was reported that T3/T4 ratio is also useful as a prognostic value for the relapse of hyperthyroidism in post-treated hyperthyroid subjects and in the differentiation of atypical hyperthyroidism from destructive thyroiditis [23,24]. This study seeks to compare T3/T4 ratio in obese and lean children and adolescents, and correlates thyroid hormones with BMI in obese Nigerian children and adolescents.

## 2. Patients and Methods

This is a retrospective study of records of 76 obese children and adolescents who were referred for evaluation because of their obese status at Aminu Kano Teaching Hospital, Kano, Nigeria. The study was approved by ethics committee of Aminu Kano Teaching Hospital (No. AKTH/MAC/SUB/12/1/P3, 24 November 2005). These children and adolescents were referred to the Chemical Pathology Laboratory for thyroid hormones determination after physical examination by the attending physicians. The clinical and demographic findings were obtained from the medical record of the subjects. They were 26 boys (mean age: 10.9 ± 0.35 years; range: 7.3–14.5 years) and 50 girls (mean age: 10.8 ± 0.4 years; range: 5.2–16.4 years). They had a mean BMI of 31.7 ± 0.1 kg/m^2^ (range: 30.2–33.6 kg/m^2^). The control subjects consisted of 20 age-matched non-obese ambulatory children and adolescents, with a mean age of 11.0 ± 0.47 years (range: 6.8–15.2 years) and a BMI of 20.2 ± 0.20 kg/m^2^ (range: 18.4–21 kg/m^2^). The control subjects were individuals who reported for routine medical check for the purpose of obtaining certificate of fitness. The clinical and demographic findings were also obtained from medical records, and measurements of weight and height were taken by qualified nurses. Obesity was defined as BMIs ≥95th percentile for children and adolescents of the same age and gender according to Center for Disease Control (CDC) pediatric growth charts. The BMI for age–weight categories and their corresponding percentiles are as follows: underweight, <5th percentile; normal, >5th but <85th percentile; overweight, >85th but <95th percentile; obese, ≥95th percentile; severely obese, >99th percentile. Adolescence was defined as a period of life when a child develops from puberty to maturity. 

### 2.1. Sample Collection and Preparation

Five milliliters of venous blood were obtained in a fasting state into plain tube and allowed to clot for 30 min. The blood was centrifuged at 3000 rpm for 10 min and serum separated. The serum was stored at −20 °C until analyses were performed.

#### 2.1.1. Analytical Methods

Triiodothyronine, thyroxine, and thyroid stimulating hormone were determined using an ELECSYS 1010 auto-analyzer (Roche Diagnostics, Penzberg, Germany). The technique employed the principle of chemiluminescence immunoassay. The T3/T4 ratio was calculated by dividing the concentration of T3 by T4 value in the same unit. The reference ranges for our laboratory are T3 = 1.3–3.1 nmol/L, T4 = 66–181 nmol/L, and TSH = 0.27–4.0 µIU/mL.

#### 2.1.2. Statistical Analysis

Statistical analysis was performed using SPSS 20 version (IBM, Armonk, NY, USA), and an unpaired Student’s *t*-test was used to compare the mean levels of thyroid hormones between obese and non-obese subjects, while multiple regression analysis was used to correlate each thyroid hormone, T3/T4 ratio, and BMI among obese and non-obese subjects. Values of *p* < 0.05 were regarded as statistically significant.

## 3. Results

The results are expressed as mean ± standard errors of means because of the small sample size. Table 1 shows the comparison of measured parameters between obese and non-obese children and adolescents. BMI (*p* < 0.001), T3 (*p* < 0.01), TSH (*p* < 0.001), and T3/T4 ratio (*p* < 0.001) were significantly higher in obese than non-obese children and adolescents. 

Triiodothyronine (*r* = 0.230; *p* < 0.05), TSH (*r* = 0.272; *p* < 0.02), and T3/T4 ratio (*r* = 0.232; *p* < 0.05) correlated positively with BMI, while T3 (*r* = 0.235; *p* < 0.05) and T3/T4 ratio (*r* = 0.231; *p* < 0.05) also correlated positively with diastolic blood pressure in obese children and adolescents. The correlation between thyroid hormones and systolic/diastolic blood pressure was insignificant (Table 2). 

Table 3 shows the characteristics of measured variables in obese boys compared with obese girls. Even though the measured parameters were higher in boys than girls, only T3/T4 ratio (*p* < 0.005) was statistically significant.

Table 4 presents the comparison of measured parameters in obese subjects based on the levels of TSH. The values of measured parameters in those with TSH levels > 4.0 µIU/mL (which is the upper limit of reference range) were not significantly different compared to their counterparts with TSH < 4.1 µIU/mL. 

Table 5 indicated the correlation of TSH with other measured thyroid hormones in obese and non-obese children and adolescents. Triiodothyronine (*r* = 0.241; *p* < 0.05) and T3/T4 ratio (*r* = 0.274; *p* < 0.002) correlated positively with TSH in obese children, while an insignificant correlation was observed between TSH and other thyroid hormones in non-obese children/adolescents.

## 4. Discussion

This study observed a positive association between T3, TSH, T3/T4 ratio, and BMI in obese children and adolescents. This is consistent with previous studies [9,25,26,27,28,29,30,31,32,33], wherein positive associations between TSH and BMI [25], and between weight gain and progressive increase in TSH levels was reported [9]. It has been suggested that thyroid hormones within normal reference ranges were probably contributing factors to weight gain in the general population [25]. On the other hand, some authors have suggested that obesity by itself may be the cause of the observed changes in thyroid hormones [26]. Solanki et al. [32] and other authors [33] reported a significant positive association between participants’ BMI and their TSH mean serum levels, which was stated to suggest a state of possible sub-clinical hypothyroidism [15]. The association between TSH and BMI has been said to be under the influence of adipose tissue signals, and leptin has also been said to possibly have a significant effect on the central regulation of thyroid function through thyrotropin releasing hormone [34]. 

Significantly higher levels of T3, TSH, and T3/T4 ratio were observed among obese children and adolescents compared with non-obese individuals. This is consistent with reports on obese German children [26]. Moderately higher TSH levels with normal or slightly higher free T4 (fT4) and/or free T3 (fT3) levels were reported in obese subjects, which may suggest an adaptation process to increase energy expenditure [26]. However, we did not measure fT3 and fT4 in this study as they are not routinely assayed in our laboratory. The high prevalence of elevated TSH compared with controls as reported in the cross-sectional study of obese German children was accompanied with significantly higher T3 levels, albeit within the normal reference range. The data presented in our study revealed significantly higher T3 levels in obese children and adolescents compared with non-obese subjects. Although the mean T3 level was within the normal reference range (1.3–3.1 nmol/L), it is possible that the relatively higher T3 levels may have impacted the body weight. This may have been responsible for the significantly higher T3/T4 ratio observed in obese children, which may be an indication of thyroid function and thyroid hormone action on tissues. Triiodothyronine plays very important and critical roles that influence body weight, and such roles include temperature homeostasis, resting energy expenditure (REE) by modulating the interaction with adipose tissue to influence the adaptations of metabolism, spontaneous motor activity and thermogenesis [34]. The T3/T4 ratio can be elevated in hyperthyroidism and hypothyroidism due to higher efficiency of the thyroid gland in secreting T3 as well as increased 5′ deiodinase tissue activity [26]. It was reported that T3/T4 ratio may be higher due to the decline in T4 levels. We observed, compared to non-obese children, an insignificantly lower mean level of T4 in obese children in our study. Some authors who have evaluated T3/T4 ratio in euthyroid subjects suggested that the ratio may be used as an additional parameter in the early recognition of future thyroid dysfunction [22]. Clinical studies in children have indicated that 7–23% of obese children exhibit elevated TSH levels [35], but in our study only 7 out of 76 (9.2%) had TSH levels above the normal reference range. 

Our data demonstrated that there were gender differences between obese boys and girls when the T3/T4 ratio was compared: obese boys had a significantly higher ratio than that of obese girls. Minami et al. [16] reported gender differences in the association between thyroid hormones and obesity. Significantly lower levels of TSH, fT3, and fT4 were reported in girls, while TSH and fT3 were lower among boys in a population study among Indian children [36]. However, we observed an insignificant change in the levels of measured thyroid hormones among boys and girls. Our observation is consistent with several other studies of healthy children from the United States, Canada, and Netherlands [37,38,39]. Gender differences were observed to vary by racial or regional characteristics [35]. 

A number of mechanisms have been used to explain thyroid dysfunction in obesity. The thyroid dysfunctions could serve as an adaptation process that increases energy expenditure in an attempt to reduce further weight gain [1]. The most popularly cited mechanisms are the impact of leptin, thyroid hormones resistance, mitochondrial dysfunction [40], changes in the activity of deiodinases, chronic low-grade inflammation, and insulin resistance. Some authors have suggested that leptin exerts some influence on the hypothalamic-pituitary axis, which can cause changes in thyrotropin-releasing hormone secretion [41,42,43,44,45]. Leptin is an important factor that influences food intake and energy expenditure, and its concentration had been demonstrated to be proportional to the body adiposity [1]. Thyroid stimulating hormone may directly stimulate differentiation of pre-adipocytes and the production of leptin by adipocytes via its receptors in adipose tissues [46,47,48]. The elevated TSH levels could be explained by T3 resistance resulting from an impaired negative feedback mechanism between TSH and the circulating thyroid hormones, caused by a reduced level of T3 receptors in the pituitary of obese subjects [1]. Apart from the effects of secreting products of adipocytes, thyroid hormones are important regulators of energy expenditure and appetite regulation. Body weight is regulated through a fine-tuning between energy intake and energy consumption, while energy consumption is determined by resting energy expenditure, non-exercise activity, and voluntary physical activity. In euthyroid non-obese individuals, only about 20% of the circulating T3 is secreted by the thyroid gland, while the rest T3 comes from peripheral deiodination of T4 in a reaction catalyzed by type 1 5′deiodinase that occurs in the kidney and liver, while type 2 deiodination takes place in the skeletal muscle, the heart muscle, white and brown adipocytes, and glia cells. However, in obese subjects, a higher proportion of circulating T3 may be secreted by thyroid gland and adipocytes. Obesity is characterized by chronic low-grade inflammation and may appear to be related to thyroid function [49]. Some authors have reported a positive association between the free T3/free T4 ratio and serum concentrations of interleukin-6 and high-sensitivity C-reactive protein, regardless of body weight [50], while others have observed a positive correlation between BMI and volume of the thyroid gland [51]. Inflammation can affect thyroid function by modulating the expression and/or activity of deiodinase in different body tissues [52,53,54]. However, the exact mechanisms involved in the regulation of deiodinases by chronic inflammation present in obese subjects are not known [49]. The increase in serum TSH levels may stimulate the production of inflammatory cytokines by adipocytes [49]. The cybernetic principle of integrative thyroid control was used to describe the relationship between obesity and changes in thyroid hormone levels [55]. Obesity was stated to be a consequence of type 2 allostatic load and to possibly result in adaptive responses of thyroid homeostasis. Others have shown that elevated TSH levels and increased total step-up deiodinase activity (within the normal reference range) was observed in subjects with weight gain [55]. 

We observed a significant positive correlation between TSH and T3 in obese children. This may indicate a TSH-T3 shunt in obese children. Some authors have suggested the existence of a TSH-T3 shunt in humans, which is mediated by functional thyroid tissue with the aim to compensate for abating thyroid function during the onset of hypothyroidism [56]. The relationship between TSH and T3 in non-obese children was not significant.

The limitations of this retrospective study are its small size and the inability to assay free triiodothyronine and free thyroxine routinely in our setting. In addition, step-up deiodinases activity (SPINA-GD) could not be calculated because of the non-availability of free T3 and/or free T4 values. A prospective study involving a larger population of obese children is suggested.

In conclusion, serum T3, TSH, and T3/T4 ratio, were significantly higher in obese children and adolescents compared with subjects that were non-obese. The T3/T4 ratio was higher in obese boys than girls, and 9.2% of the obese children had TSH levels above the upper limit of the normal reference range. The T3, TSH, and T3/T4 ratio correlated positively with BMI in obese Nigerian children and adolescents. Mechanisms linking hyperdeiodination to obesity are possible: obesity might result from elevated thyroid hormones, high T3-syndrome may be a consequence of obesity, and both phenomena may have a common cause in type 2 allostasis. Since thyroid dysfunction represents a continuum from asymptomatic to clinical symptomatic disease, it is suggested that obese children should be counseled on the importance of weight loss to avoid the risks associated with obesity. 

## Figures and Tables

**Table 1 medsci-05-00036-t001:** Comparison of measured thyroid hormones between obese and non-obese children and adolescents (mean ± standard error of mean). Range is shown between parentheses.

Measured Variables	Obese Children	Non-Obese Children	*p*-Value
Number of subjects	76	20	
Age (years)	10.7 ± 0.3 (5.5–15.9)	11.0 ± 0.47 (6.8–15.2)	>0.05
Blood pressure (mmHg)			
Systolic	110.3 ± 1.1 (91–129)	108 ± 1.2 (97–118)	>0.05
Diastolic	68.4 ± 0.9 (53–83)	67.6 ± 0.8 (59–74)	>0.05
Body mass index (kg/m^2^)	31.7 ± 0.10 (30.2–33.6)	20.2 ± 0.20 (18.4–21.0)	<0.001
Triiodothyronine (T3) nmol/L	2.28 ± 0.02 (1.93–2.63)	1.88 ± 0.04 (1.52–2.23)	<0.01
Thyroxine (T4) nmol/L	94.9 ± 0.42 (87–102.2)	98.5 ± 1.07 (88.9–108)	>0.05
Thyroid stimulating hormone (TSH) µIU/mL	3.30 ± 0.02 (2.95–3.65)	2.11 ± 0.06 (1.57–2.64)	<0.001
T3/T4 ratio	0.024 ± 0.001 (0.007–0.041)	0.019 ± 0.001 (0.010–0.028)	<0.001

**Table 2 medsci-05-00036-t002:** Correlation of measured thyroid hormones and blood pressure with body mass index in obese subjects.

Parameters	*r*-Value	*p*-Value
Triiodothyronine vs. body mass index	0.230	<0.05
Triiodothyronine vs. systolic blood pressure	0.216	=0.09
Triiodothyronine vs. diastolic blood pressure	0.235	<0.05
Thyroxine vs. body mass index	0.192	>0.1
Thyroxine vs. systolic blood pressure	0.217	=0.09
Thyroxine vs. diastolic blood pressure	0.194	>0.1
Thyroid stimulating hormone vs. body mass index	0.272	<0.02
Thyroid stimulating hormone vs. systolic blood pressure	0.220	=0.08
Thyroid stimulating hormone vs. diastolic blood pressure	0.218	=0.09
T3/T4 ratio vs. body mass index	0.232	<0.05
T3/T4 ratio vs. systolic blood pressure	0.225	=0.07
T3/T4 ratio vs. diastolic blood pressure	0.231	<0.05

**Table 3 medsci-05-00036-t003:** Comparison of measured variables between obese boys and girls.

Measured Parameters	Boys (*n* = 26)	Girls (*n* = 50)	Total (*N* = 76)
Age (years)	10.9 ± 0.35 (7.3–14.5)	10.8 ± 0.4 (5.2–15.9)	10.7 ± 0.3 (5.5–15.9)
Blood pressure (mmHg)			
Systolic	111.8 ± 1.2 (99–123)	108.5 ± 1.2 (91–125)	110.3 ± 1.1 (91–129)
Diastolic	69.6 ± 0.93 (60–78)	67.2 ± 1.1 (52–82)	68.4 ± 0.9 (53–83)
Body mass index (kg/m^2^)	32.5 ± 0.2 (32–33)	31.4 ± 0.25 (30–32)	31.7 ± 0.1 (30–33)
T3 (nmol/L)	2.42 ± 0.08 (1.6–3.23)	2.21 ± 0.11 (0.65–3.76)	2.28 ± 0.02 (1.93–2.63)
T4 (nmol/L)	95.2 ± 1.33 (81–109)	94.8 ± 1.56 (72–117)	94.4 ± 0.42 (87–117)
TSH (µIU/mL)	3.9 ± 0.16 (2.27–5.53)	3.0 ± 0.17 (0.6–5.5)	3.30 ± 0.02 (2.9–3.65)
T3/T4 ratio	0.028 ± 0.002 (0.0076–0.048)	0.022 ± 0.002 (0.0063–0.050)	0.024 ± 0.001 (0.007–0.050)

**Table 4 medsci-05-00036-t004:** Comparison of measured parameters in obese children and adolescents based on serum TSH levels (mean ± standard error of mean).

Parameters	TSH < 4.1 µIU/mL	TSH > 4.1 µIU/mL	*p*-Value
Numbers of subjects	69	7	
Body mass index (kg/m^2^)	31.6 ± 0.06 (30–33)	31.9 ± 0.08 (31–32)	>0.05
T3 (nmol/L)	2.28 ± 0.04 (1.6–2.9)	2.27 ± 0.08 (1.8–2.7)	>0.05
T4 (nmol/L)	94.4 ± 0.72 (82–106)	95.7 ± 3.4 (77–114)	>0.05
TSH (µIU/mL)	3.15 ± 0.01 (2.9–3.3)	4.77 ± 0.04 (4.5–4.9)	0.001
T3/T4 ratio	0.024 ± 0.001 (0.007–0.04)	0.024 ± 0.01 (0.007–0.04)	1.0

**Table 5 medsci-05-00036-t005:** Correlation of TSH with other thyroid hormone levels in obese and non-obese children/adolescents.

Parameters	*r*	*p*-Values
Obese children		
TSH/triiodothyronine	0.241	<0.05
TSH/thyroxine	0.216	=0.08
TSH/T3/T4 ratio	0.274	<0.002
Non-obese children		
TSH/triiodothyronine	0.198	=0.08
TSH/thyroxine	0.190	=1.0
TSH/T3/T4 ratio	0.196	=0.09

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
