# Peer review of "The Association of Triiodothyronine-to-Thyroxine Ratio with Body Mass Index in Obese Nigerian Children and Adolescents"

_medsci, 2017, doi:10.3390/medsci5040036_

Reviewer 1 Report

The authors are focusing on an important and interesting topic and, therefore, in my opinion, this manuscript should be published in Medical Sciences.

However, an extensive revision is absolutely necessary:

1) There is no sufficient information concerning the participants. What was the age range of the children and adolescents? How was adolescence defined? Why did the children and adolescents contact the hospital? Is there any information reagrding medical history?

How were overweight and obesity defined? It is not recommended to use the BMI cuttoffs for adults, nor for subadults. Percentiles should be used for each age and sex separately. 

How was BMI determine? Were all participants measured by trained investigators?

How did recruitment take place?

According to Table 1, non-obese children were significantly older than obese ones, so in this case, the control-group is not age-matched. Please provide the age range of obese participants and non-obese controls.

2) The presentation of the results is quite poor. In tables 1, 3, and 4, ranges should be presented.

3) I would suggest a regression analyses to test the impact of hormonal parametes on BMI.

Author Response

1) Additional information such as blood pressure, who did physical and clinical examination and measured the weight and height is included. The age range and definition of adolescent included.

Overweight/obesity is defined and obesity is defined in percentile.

Weight and height were measured by qualified nurses.

Thank you, age range was provided in Table 1.

2) Ranges were provided.

3) Multiple regressions were done.

Reviewer 2 Report

In their manuscript titled "Association of Triiodothyronine to thyroxine ratio with body mass index in obese Nigerian children and adolescents”, Mathias Abiodun Emokpae and Obazelu Arhenrhen Progress report associations of TSH and T3 concentrations as well as T3/T4 ratio with obesity in Nigerian children. Additionally, they describe parameters of thyroid homeostasis to correlate to body mass index.

This is an interesting and well-conducted study that addresses an important question. However, due to some minor flaws in reporting and some omissions, acceptance cannot be considered in the current state.

1) First of all, the validity of T3/T4 ratio has recently been questioned. In order to make the results less ambiguous the authors should also calculate step-up deiodinase activity (SPINA-GD) from T3 and T4 concentrations and report the results in the text and the tables.

2) Table 4 seems to be somewhat artificial, since the validity of the cut-off value of 4.1 µIU/ml for TSH concentration is at question. It would be useful to complement table 4 with another table that reports correlations of TSH concentration to T3 and T4 concentrations as well as to T3/T4 ratio and SPINA-GD. Recent research revealed the existence of a functional TSH-T3 shunt in humans, and it would be interesting to learn about this relation in children. Correlations should be reported separately for obese children, non-obese children and for the total population.

3) The relation between obesity and thyroid hormones has recently received considerable attention. According to newer theories obesity may be regarded as a consequence of type 2 allostasis in the setting of chronic psychosocial stress (at least in a significant subgroup of obese patients), and it could also be shown that type 2 allostasis is associated to elevated concentrations of TSH and peripheral thyroid hormones. Therefore, obesity, step-up hyperdeiodination and elevated concentrations of thyroid hormones may have a common cause. The authors might want to comment on this interesting new perspective in their discussion. See e.g. review articles by Ghike (PMID 27876129), Chatzitomaris et al. (PMID 28775711) and an excellent recent paper by Fontenelle et al. (PMID 27923249) for reference. Additionally, the recent discovery of the TSH-T3 shunt should be commented (see publications by Josef Koehrle et al. and Rudolf Hoermann et al. for reference). In summary, the authors should point out that three mechanisms linking hyperdeiodination to obesity are possible: Obesity might result from elevated thyroid hormones, high T3-syndrome may be a consequence of obesity and both phenomena may have a common cause in type 2 allostasis.

4) The reviewer assumes that additional clinical parameters (e.g. blood pressure, pulse frequency, lipid concentrations, glycated hemoglobin and waist circumference) have been obtained in the included subjects. If this is the case, the manuscript could be upvalued by calculating a score for allostatic load (see a recent research paper by Robertson et al, PMID 28813505, for an example, how to calculate this score). If possible, the authors should report correlation of the allostatic load score with age and parameters of thyroid homeostasis.

5) Minor issues include the following points:

Line 47: “impair” should be replaced by “impaired”.

Line 48: “increase” should be replaced by “increased”.

Tables 1, 2 and 3: Although in the text p values are reported to be <0.001, <0.01 and so on, the corresponding p values are denoted in the table as exactly being 0.001, 0.01 etc. The reviewer recommends to report the values with the less sign and, to be more in line with usual notation, to report non-significant results as “n.s.” rather than “>0.05”. Please also report the results for SPINA-GD in the two groups and the corresponding alpha error in all three tables.

Line 174 and 175: It is unclear, what the authors want to say with the term “fine-turning”. Possibly, they wanted to write “fine-tuning”.

Line 178: The term “1,5deiodinase” is unknown to the reviewer. Did the authors want to write “type 1 5’-deiodinase”? Additionally, the kidney isn’t the only extrathyroidal place of step-up deiodination (and it isn’t even the most important). Type 1 deiodination also takes place in the liver and type 2 deiodination in skeletal muscle, heart muscle white and brown adipose tissue, glia cells and other organs. This should be mentioned here.

Line 184: The reviewer thinks that “boys had girls” should be replaced by “boys than girls”.

Author Response

1) We could not calculate step-up deiodinases activity (SPINA-GD) because of non-availability of free T3 or free T4 values.

2) Table 5 showing correlation of TSH with other thyroid hormone levels is included. A statement describing the existence of TSH-T3 shunt in the light of our finding is included.

3) The references suggested above were very helpful. Thank you very much for providing them. We made good use of them in the discussion section. Please find additions in red ink.

4) We however could not calculate the allostatic load because of limited information.

5) Corrections made in the manuscript.

Round  2

Reviewer 1 Report

The manuscript is now suitable for publication

Reviewer 2 Report

"SPINAD-GD" in line 219 should be replaced by "SPINA-GD". Likewise, "Frontenelle" in line 349 should be replaced by "Fontenelle".

Otherwise, all major issues raised by the reviewer have been addressed (with the exception of some points that cannot be changed due to missing data). Acceptance may be considered now.